# Tackling syndemics by integrating infectious and noncommunicable diseases in health systems of low- and middle-income countries: A narrative systematic review

**Angela Jackson-Morris**[1]*, **Sarah Masyuko**[1,2], **Lillian Morrell**[1,3], **Ishu Kataria**[1], **Erica L. Kocher**[1,4], **Rachel Nugent**[1]

1 Center for Global Noncommunicable Diseases, RTI International, Research Triangle Park, North Carolina, United States of America, 2 Department of Global Health, University of Washington, Seattle, Washington, United States of America, 3 Wilson Sheehan Lab for Economic Opportunities, University of Notre Dame, Notre Dame, Indiana, United States of America, 4 Emory University, Emory University, Atlanta, Georgia, United States of America

* ajackson-morris@rti.org

## Abstract

The co-occurrence of infectious diseases (ID) and non-communicable diseases (NCD) is widespread, presenting health service delivery challenges especially in low-and middle-income countries (LMICs). Integrated health care is a possible solution but may require a paradigm shift to be successfully implemented. This literature review identifies integrated care examples among selected ID and NCD dyads. We searched PubMed, PsycINFO, Cochrane Library, CINAHL, Web of Science, EMBASE, Global Health Database, and selected clinical trials registries. Eligible studies were published between 2010 and December 2022, available in English, and report health service delivery programs or policies for the selected disease dyads in LMICs. We identified 111 studies that met the inclusion criteria, including 56 on tuberculosis and diabetes integration, 46 on health system adaptations to treat COVID-19 and cardiometabolic diseases, and 9 on COVID-19, diabetes, and tuberculosis screening. Prior to the COVID-19 pandemic, most studies on diabetes—tuberculosis integration focused on clinical service delivery screening. By far the most reported health system outcomes across all studies related to health service delivery (n = 72), and 19 addressed health workforce. Outcomes related to health information systems (n = 5), leadership and governance (n = 3), health financing (n = 2), and essential medicines (n = 4)) were sparse. Telemedicine service delivery was the most common adaptation described in studies on COVID-19 and either cardiometabolic diseases or diabetes and tuberculosis. ID-NCD integration is being explored by health systems to deal with increasingly complex health needs, including comorbidities. High excess mortality from COVID-19 associated with NCD-related comorbidity prompted calls for more integrated ID-NCD surveillance and solutions. Evidence of clinical integration of health service delivery and workforce has grown–especially for HIV and NCDs —but other health system building blocks, particularly access to essential medicines, health financing, and leadership and governance, remain in disease silos.

**Data Availability Statement:** Relevant data pertaining to the search terms, results, and the studies reviewed is included in the research article.

**Funding:** This work was funded via an internal funding award from RTI International for research and development to RA and EE. The funders had no role in study design, data collection and analysis, decision to publish, or preparation of the manuscript.

**Competing interests:** The authors have declared that no competing interests exist.

## Introduction

Co-occurrence of infectious diseases (IDs) and non-communicable diseases (NCD) in patients is more common than in the past as demographic and epidemiological transitions affect most countries. More patients with comorbidities call upon health systems to offer a wider service package, gather additional data, and train health professionals to detect and manage complex disease combinations [1]. While diseases differ in their pathogenesis, some IDs and NCDs share patient or risk factors and hence can be managed in similar ways [2]. In addition, the underlying socio-economic determinants and health disparities may be similar despite differing causes. COVID-19 is becoming the iconic example of an ID that especially affects people living with chronic disease. A nascent literature on syndemic disease—defined as clusters of two or more socially-driven diseases, the interaction of which yields worse health outcomes than those diseases individually—highlights the need to understand comorbid disease through a public policy lens [3–6].

Some limited evidence shows the potential for health systems to become more integrative across diseases and risks, yet few examples outside the HIV context are well documented, especially in relation to national priorities and across different types of health systems [7]. LMIC health system managers are seeking a paradigm shift to design and adopt chronic care models, integrate essential medicine supply chains, and integrate health information systems, while leveraging the existing infrastructure and health workforce that has little training in chronic disease. Integrated ID-NCD health care is poorly defined, and many "models" have been described. Most experience of integrated ID-NCD care is for people living with HIV with comorbid chronic conditions and examples are largely limited to delivery of care. There are limited published examples of NCD integration into primary care and these are often small-scale, and there is a dearth of evidence about how to design and scale up ID-NCD interventions.

This systematic review describes the ID-NCD comorbid health burden in LMICs, analyzes published examples of ID-NCD integrated care, and rates the quality of the integration evidence for its effect on health outcomes, for a range of prevalent disease dyads.

## Materials and methods

We searched for relevant literature reporting intervention evaluations on integrated healthcare for infectious and non-communicable diseases in LMICs. These included, but were not limited to formative/qualitative studies, case studies, pilot evaluations, uncontrolled evaluations, quasi-experimental evaluations, RCTs, economic evaluations, and policy analyses. We defined integration using World Health Organization terminology—"the management and delivery of health services so that clients receive a continuum of preventive and curative services, according to their needs over time and across different levels of the health system" [8].

We reviewed studies that describe integration of selected infectious diseases [tuberculosis, neglected tropical diseases (onchocerciasis and trachoma), and COVID-19], and non-communicable diseases [diabetes, obesity, and cardiovascular disease]. Between these two categories, we searched for the disease dyads: (a) tuberculosis + diabetes; (b) tuberculosis + diabetes + COVID-19; (c) onchocerciasis/trachoma + diabetes; (d) onchocerciasis/trachoma + diabetes + COVID-19; and (e) COVID-19 + obesity/ diabetes / cardiovascular diseases. We did not search for integration evidence that included HIV and AIDS, as there is plentiful literature, including, for example, an entire special edition of the Journal of AIDS and a multi-country portfolio of NIH research projects (e.g. NHLBI-SIMPLE) [9–15]. Therefore we focused our inquiry on NCD co-morbidities that have received less attention. Although we did not expect to find many studies, we searched for evidence of integration between NCDs and neglected

**Table 1. Summary of health system and patient health outcomes for integrated care.**

| Outcome | Performance measurement |
|---|---|
| **System outcomes (WHO building blocks)** | |
| **Health Service Delivery** | • Efficiency: Unnecessary duplication of tests; Number of consultations per doctor <br> • Adaptability: Introduction of new models of care to meet emerging expectations <br> • Coverage: Schedule of available funded procedures and treatments; Patient reported confidence in ability to access care. Consequences of unmet need <br> • Healthcare processes: Models of care; Patient pathways and protocols; Coordination and integration processes; Flow of information; Collaboration <br> • Care coordination: joint needs assessment, joint care planning, joint care management and joint discharge planning. <br> • Cost of service delivery |
| **Health Workforce** | • Joint training <br> • Multidisciplinary teams/ Task shifting/task sharing /role revision |
| **Health Information** | • Integrated health information system (manual/electronic) |
| **Essential Medicines** | • Joint/ pooled procurement: Also known as group purchasing, WHO defines pooled procurement as "Purchasing done by one procurement office on behalf of a group of facilities, health systems or countries" [17]. <br> • Integrated supply chain and logistics management information systems: integrating processes across diseases |
| **Health Financing** | • Financing models: Pooled or aligned resources <br> • Universal health care <br> • Utilisation of cost-effective alternative models of care. <br> • Assured supply of essential drugs |
| **Leadership and governance** | • Joint policies/universal health care |
| **Patient outcomes** | |
| **Patient reported outcomes** | • Person-centered care <br> • Improved patient experiences <br> • Patient satisfaction |
| **Value and sustainability** | • Care is provided in the right place at the right time <br> • Demand is well managed <br> • Sustainable fit between needs and resources |

tropical diseases (NTDs) due to NTD endemicity in many LMICs and strong policy attention upon eradication and/or containment goals [16]. While recognizing malaria as a high impact condition, we opted not to prioritize this, given the review focus on health system building blocks whereas many malaria interventions are outside the health system. Recognizing the recent attention to interactions between epidemic COVID-19 and NCDs, we sought lessons learned from integrating endemic IDs with NCDs as a counterpart, but no literature was identified concerning this dyad. The evidence we found was concentrated on the dyads of tuberculosis and diabetes, and comorbid COVID-19 with multiple NCDs. We concentrated on relevant interventions across the WHO health system building blocks for adults in LMICs living with ID and NCDs. Our comparators were the health system functions that address only a single disease or a group of closely related diseases (i.e. nonintegrated). We extracted outcomes that reflect indicators of integrated care across the major health system functions as defined by WHO (Table 1).

To analyze and present the review results, we prepared tree maps of Health System Outcomes identified in the literature as per the WHO building blocks. These maps were created using Excel and show the number of articles represented under each health system outcome.

## Inclusion and exclusion criteria

We included studies that were (a) published in peer reviewed literature up to December 2022; (b) in English; (c) report quantitative or qualitative data on health service delivery programs,

policies, or functions that include a combination of tuberculosis, diabetes, cardiovascular disease, hypertension, onchocerciasis, trachoma, or COVID-19; (d) conducted in low-and middle-income countries as per the World Bank Income groupings for 2021; and (e) describe integration, integrated care, multi-morbidity, or dual diagnosis.

## Search strategy

We searched the following electronic databases: PubMed, PsycINFO, Cochrane Library, CINAHL, Web of Science, EMBASE, and Global Health Database. In addition, we conducted secondary reference searching on all studies included in the review. We also searched for ongoing randomized clinical trials through clinicaltrials.gov, the WHO International Clinical Trials Registry (ICTRP), Pan African Clinical Trials Registry (PACTR), and the Australian New Zealand Clinical Trials Registry to gather evidence. Our search included two components: (a) disease component and (b) integration component as per the dyads, adapting search terms as needed for each electronic database (Box 1).

---

### Box 1. Search strategy

**Full search in PubMed format**

**Search for Tuberculosis and Diabetes integration**

[("Tuberculosis"[MeSH] OR TB[tw] OR tuberculosis[tw])

AND

("Diabetes Mellitus"[Mesh] OR diabetes[tw] OR diabetic[tw] OR iddm[tw] OR niddm [tw])

AND

("Delivery of Health Care, Integrated"[MeSH] OR "systems integration"[MeSH] OR integrat*[tw] OR

(("Multimorbidity"[Mesh] OR "multi morbid*"[tw] OR multimorbid*[tw] OR "dual diagnoses"[tw] OR syndemic*[tw] OR comorbid*[tw] OR "co morbid*"[tw]) AND ("Delivery of Health Care"[Mesh] OR "healthcare system*"[tw] OR "health care system*"[tw] OR "healthcare delivery"[tw] OR "health care delivery"[tw] OR "service delivery"[tw] OR "continuity of care"[tw] OR "health service*"[tw] OR "health system*"[tw] OR "health polic*"[tw] OR "health care polic*"[tw] OR "healthcare polic*"[tw] OR "healthcare management"[tw] OR "health care management"[tw])))]

AND (See location list below)

**Search for COVID-19, Obesity, Cardiovascular diseases and Diabetes integration**

("COVID-19"[MeSH] OR SARS-CoV-2[MeSH] OR COVID-19[tw] OR COVID-19[tw] OR "coronavirus 2019"[tw] OR "2019 nCOV"[tw] OR "SARS-CoV-2"[tw] OR "SARS CoV2"[tw]) AND ("Diabetes Mellitus"[Mesh] OR diabetes[tw] OR diabetic[tw] OR iddm[tw] OR niddm[tw] OR "Obesity"[MeSH] OR obes*[tw] OR "Cardiovascular diseases"[MeSH] OR CVD[tw] OR "cardiovascular disease*"[tw] OR "heart disease*"[tw])

AND

---

("adapt*"[tw] OR "modif*"[tw] OR "healthcare system*"[tw] OR "health care system*"[tw] OR "healthcare delivery"[tw] OR "health care delivery"[tw] OR "service delivery"[tw] OR "continuity of care"[tw] OR "health service*"[tw] OR "health system*"[tw] OR "health polic*"[tw] OR "health care polic*"[tw] OR "healthcare polic*"[tw] OR "healthcare management"[tw] OR "health care management"[tw])

AND (See location list below)

**Search for COVID-19, Tuberculosis and Diabetes integration**

(((((((((((((((((Tuberculosis[MeSH] OR TB[All Fields] OR tuberculosis[All Fields]))) AND ((diabetes[MeSH] OR diabetes mellitus[MeSH] OR diabetes[All Fields] OR diabetic[All Fields] OR type 1 diabetes[All Fields] OR type 2 diabetes[All Fields]))) AND ((delivery of Health Care, Integrated[MeSH] OR systems integration[MeSH] OR integrated health care system[tw] OR integrated[tw] OR integrated health care systems[All Fields]))))))))))

AND (See location list below)

**Location list**

(Africa[tw] OR Asia[tw] OR Caribbean[tw] OR "West Indies"[tw] OR "South America"[tw] OR "Latin America"[tw] OR "Central America"[tw] OR "Middle East"[tw] OR "Eastern Europe"[tw] OR Oceania[tw] OR Abkhazia[tw] OR Afghanistan[tw] OR Albania[tw] OR Algeria[tw] OR Angola[tw] OR Antigua[tw] OR Barbuda[tw] OR Argentina[tw] OR Armenia[tw] OR Armenian[tw] OR Artsakh[tw] OR Aruba[tw] OR Azerbaijan[tw] OR Bahamas[tw] OR Bangladesh[tw] OR Barbados[tw] OR Benin[tw] OR Byelarus[tw] OR Byelorussian[tw] OR Belarus[tw] OR Belorussian[tw] OR Belorussia[tw] OR Belize[tw] OR Bermuda[tw] OR Bhutan[tw] OR Bolivia[tw] OR Borneo[tw] OR Bosnia[tw] OR Herzegovina[tw] OR Hercegovina[tw] OR Botswana[tw] OR Brasil[tw] OR Brazil[tw] OR Bulgaria[tw] OR "Burkina Faso"[tw] OR "Burkina Fasso"[tw] OR "Upper Volta"[tw] OR Burundi[tw] OR Urundi[tw] OR Cambodia[tw] OR "Khmer Republic"[tw] OR Kampuchea[tw] OR Cameroon[tw] OR Cameroons[tw] OR Cameron[tw] OR "Cape Verde"[tw] OR "Cabo Verde"[tw] OR "Central African Republic"[tw] OR Chad[tw] OR Tchad[tw] OR Chile[tw] OR China[tw] OR Colombia[tw] OR Comoros[tw] OR "Comoro Islands"[tw] OR Comores[tw] OR Congo[tw] OR DRC[tw] OR "Congo-Brazzaville"[tw] OR "Congo-Kinshasa"[tw] OR Zaire[tw] OR "Cote d'Ivoire"[tw] OR "Ivory Coast"[tw] OR Croatia[tw] OR Cuba[tw] OR Djibouti[tw] OR "French Somaliland"[tw] OR Dominica[tw] OR "Dominican Republic"[tw] OR "East Timor"[tw] OR "Timor Leste"[tw] OR "Timor-Leste"[tw] OR Ecuador[tw] OR Egypt[tw] OR "United Arab Republic"[tw] OR "El Salvador"[tw] OR Eritrea[tw] OR Ethiopia[tw] OR Fiji[tw] OR Gabon[tw] OR "Gabonese Republic"[tw] OR Gambia[tw] OR Gaza[tw] OR Georgia[tw] OR Georgian[tw] OR Ghana[tw] OR "Gold Coast"[tw] OR Grenada[tw] OR Guatemala[tw] OR Guinea[tw] OR Guiana[tw] OR Guyana[tw] OR Haiti[tw] OR Honduras[tw] OR India[tw] OR Maldives[tw] OR Indonesia[tw] OR Iran[tw] OR Iraq[tw] OR Jamaica[tw] OR Jordan[tw] OR Kazakhstan[tw] OR Kazakh[tw] OR Kenya[tw] OR Kiribati[tw] OR Korea[tw] OR DPRK[tw] OR Kosovo[tw] OR Kyrgyzstan[tw] OR Kirghizia[tw] OR "Kyrgyz Republic"[tw] OR Kirghiz[tw] OR Kirgizstan[tw] OR "Lao PDR"[tw] OR Laos[tw] OR Lebanon[tw] OR Lesotho[tw] OR Basutoland[tw] OR Liberia[tw] OR Libya[tw] OR Macedonia[tw] OR FYROM[tw] OR Macao[tw] OR Madagascar[tw] OR "Malagasy Republic"[tw] OR Malaysia[tw] OR Malaya[tw] OR Malay[tw]

OR Sabah[tw] OR Sarawak[tw] OR Malawi[tw] OR Nyasaland[tw] OR Mali[tw] OR "Marshall Islands"[tw] OR Mauritania[tw] OR Mauritius[tw] OR "Agalega Islands"[tw] OR Mexico[tw] OR Micronesia[tw] OR Moldova[tw] OR Moldovia[tw] OR Moldovian[tw] OR Mongolia[tw] OR Montenegro[tw] OR Morocco[tw] OR Ifni[tw] OR Mozambique[tw] OR Myanmar[tw] OR Myanma[tw] OR Burma[tw] OR Namibia[tw] OR Nauru[tw] OR Nepal[tw] OR Nicaragua[tw] OR Niger[tw] OR Nigeria[tw] OR Niue[tw] OR Pakistan[tw] OR Palau[tw] OR Palestine[tw] OR Panama[tw] OR Paraguay[tw] OR Peru[tw] OR Philippines[tw] OR Philipines[tw] OR Philipines[tw] OR Philippines[tw] OR Polynesia[tw] OR Romania[tw] OR Rumania[tw] OR Roumania[tw] OR Russia[tw] OR Russian[tw] OR Rwanda[tw] OR Ruanda[tw] OR "Saint Kitts"[tw] OR "St Kitts"[tw] OR Nevis[tw] OR "Saint Lucia"[tw] OR "St Lucia"[tw] OR "Saint Vincent"[tw] OR "St Vincent"[tw] OR Grenadines[tw] OR Samoa[tw] OR "Samoan Islands"[tw] OR "Sao Tome"[tw] OR Principe[tw] OR Senegal[tw] OR Serbia[tw] OR Montenegro[tw] OR "Sierra Leone"[tw] OR "Sri Lanka"[tw] OR Ceylon[tw] OR "Solomon Islands"[tw] OR Somalia[tw] OR Somaliland[tw] OR "South Africa"[tw] OR "South Ossetia"[tw] OR Sudan[tw] OR Suriname[tw] OR Surinam[tw] OR Swaziland[tw] OR Eswatini[tw] OR Syria[tw] OR Tajikistan[tw] OR Tadzhikistan[tw] OR Tadjikistan[tw] OR Tadzhik[tw] OR Tanzania[tw] OR Thailand[tw] OR Tibet[tw] OR Togo[tw] OR "Togolese Republic"[tw] OR Tokelau[tw] OR Tonga[tw] OR Transnistria[tw] OR Trinidad[tw] OR Tobago[tw] OR Tunisia[tw] OR Turkey[tw] OR Turkmenistan[tw] OR Turkmen[tw] OR Tuvalu[tw] OR Uganda[tw] OR Ukraine[tw] OR Uruguay[tw] OR USSR[tw] OR "Soviet Union"[tw] OR "Union of Soviet Socialist Republics"[tw] OR Uzbekistan[tw] OR Uzbek[tw] OR Vanuatu[tw] OR "New Hebrides"[tw] OR Venezuela[tw] OR Vietnam[tw] OR "Viet Nam"[tw] OR "Mekong valley"[tw] OR "Mekong delta"[tw] OR "Western Sahara"[tw] OR Sahrawi[tw] OR "West Bank"[tw] OR Yemen[tw] OR Yugoslavia[tw] OR Zambia[tw] OR Zimbabwe[tw] OR Zanzibar[tw] OR Rhodesia[tw] OR "Developing Countries"[Mesh] OR LMIC OR LMICS OR LEDC OR "less developed country" OR "least developed countries" OR "newly industrialized countries" OR "emerging markets" OR "poor countries" OR "poor country" OR "underdeveloped country" OR "low-income country" OR "low-income countries" OR "middle-income country" OR "middle-income countries" OR "low- and middle-income countries" OR "developing country" OR "developing world" OR "third world" OR "less developed countries" OR "developing nations" OR "low GDP" OR "low HDI" OR "transitional economies" OR "Global South")

## Screening abstracts, data extraction and management

We searched each database and exported the results to a common Endnote database, after removing duplicates. All references were then exported into an excel worksheet with titles and authors. To aid in the title/abstract and full-text screening, we developed title/abstract screening guidance and a full-text eligibility assessment form. In addition, we conducted an initial calibration exercise to ensure that screening was standardized. We used Rayyan Qatar Computing Research Institute (Rayyan RQCRI) software to manage retrieved studies, remove duplicate reports of the same study, and manage the title and abstract screening process. Four independent reviewers (SM, EK, IK, and LM) screened the titles, abstracts, citation information, and citation descriptor terms to identify full-text articles for further review based on inclusion criteria. Any differences that emerged were resolved through consensus and discussion with a third reviewer when necessary.

Four reviewers (SM, EK, IK, and LM) independently extracted data using a standardized form. We resolved differences through consensus and referral to a senior researcher when necessary. We extracted data on the study parameters; its description including intervention and model of integration; any additional intervention components; study design; sample size; follow-up periods and loss to follow-up, and outcomes. Data was summarized in tables and figures and assessed for commonalities.

## Critical appraisal of evidence

Reviewers assessed the methodological quality of included studies using the Joanna Briggs Institute standardized critical appraisal instrument for prevalence studies, analytical cross-sectional studies, cohort studies, diagnostic test accuracy studies, quasi-experimental studies, qualitative studies, and text and opinion studies [18]. If the answer to any checklist item was no, unclear or not applicable, it was assigned a score of 0. To standardize across studies with different extractable data, we used percent scores ranging from 0 to 100%. Total quality scores of <40%, 40–80%, >80% were regarded as low, moderate, and high quality, respectively. Disagreements were settled by discussion.

## Results

Our search identified a total of 794 studies on tuberculosis and diabetes integration, 1,797 studies for system adaptations on COVID-19 and diabetes, obesity and cardiovascular diseases, and 84 studies on tuberculosis, diabetes, and COVID-19 integration as presented below (Fig 1). We did not locate eligible studies for the other disease dyads. To note, some studies reported multiple integration characteristics and outcomes, whereas not all studies reported on each aspect of integration considered by the review (Fig 2).

The world map was created and edited in Microsoft Excel for Windows, version 16.83. The public domain link to the map base layer that was used to create the figure is available at: https://commons.wikimedia.org/wiki/File:BlankMap-World.svg.

### TB and diabetes integration

**Search results.**   Of the 794 studies identified from PubMed, Embase, Web of Science, PsychInfo and CINAHL, 286 duplicates were removed. After title and abstract screening, 115 studies proceeded to full-text review. 56 studies were eligible and included in this review.

This review included studies from 19 countries from all WHO regions except the European region. South-East Asia studies were most numerous, particularly studies in India. 4 (7%) articles were from low-income countries, 39 (70%) lower middle-income countries, and 9 (16%) from upper middle- income countries. Four multi-country studies from low income, lower-middle income countries, upper-middle income countries were identified. Table 2 presents the baseline characteristics of the included studies.

We critically appraised the methodological quality for the 56 studies that met the inclusion criteria using JBI tools, as outlined above. The majority of studies (106/111, 95.5%) of studies were moderate or high quality (Table 2). Four studies were low quality. All studies were retained in order to reflect current research on the topic.

*Summary of interventions*. Almost half of all articles (n = 26, 46%) focused on tuberculosis screening for patients receiving diabetes services. Five (9%) of these studies involved diabetic patients, and 22 (39%) involved both tuberculosis and diabetes patients. Most interventions for tuberculosis patients used either random blood glucose, fasting blood glucose, or Hemoglobin A1c (HbA1c) for diabetes screening. Among diabetes patients, interventions screened for tuberculosis using symptomatology, sputum culture, and Gene Xpert.

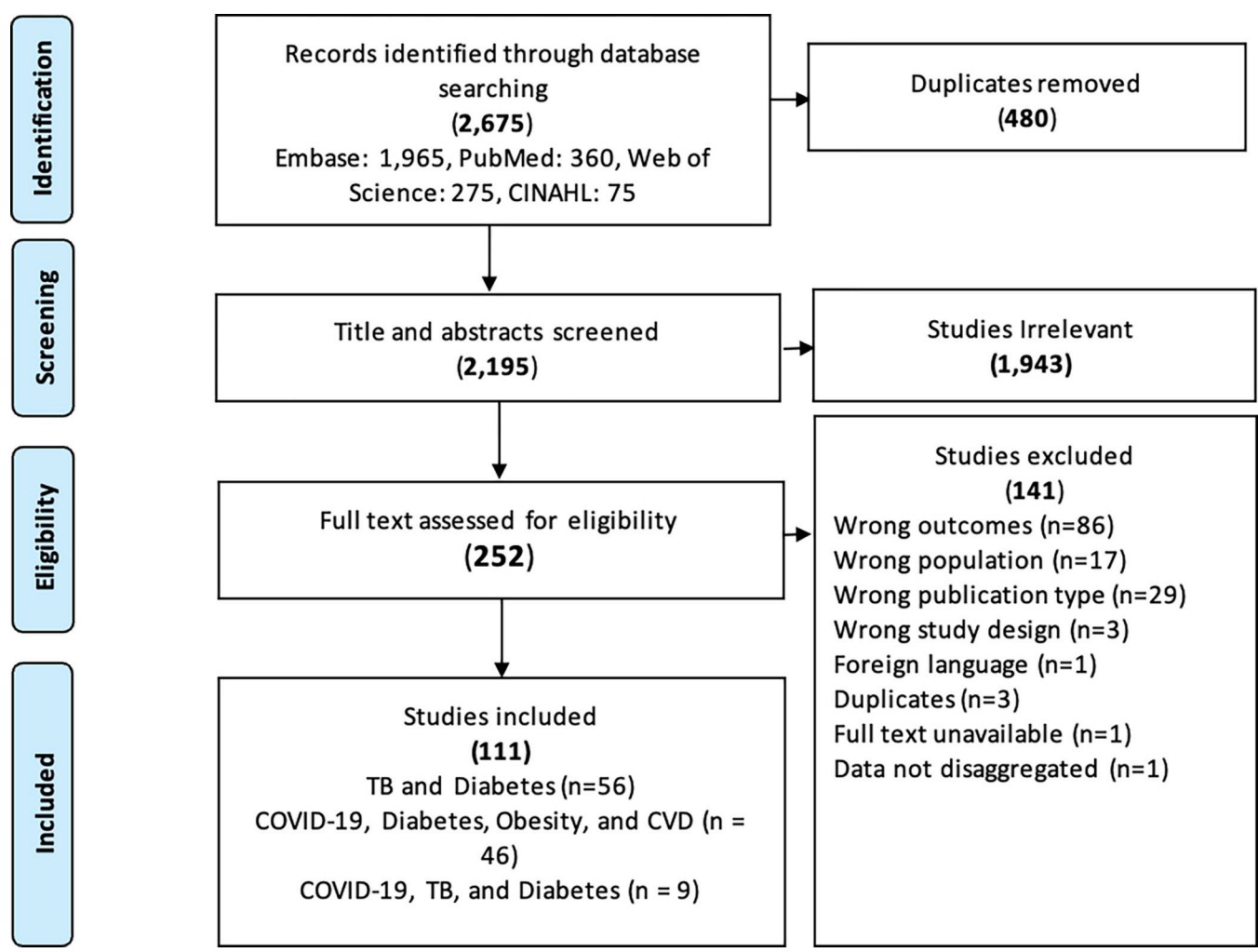

**Fig 1. Prisma flow chart of screening process of included articles.**

*Characteristics of integration.* Of the 56 articles included, 39 (70%) involved clinical integration, 11 (20%) included both clinical and professional integration, four (7%) included systems integration, and one (2%) included a mix of clinical, professional, and organizational integration. The articles all concerned population-based care. The majority (40, 71%) were concerned with micro level integration, twelve (21%) had a mix of micro and meso levels, and three (5%) were concerned with macro level integration. One study focused on service availability and a facility readiness assessment to provide NCD services. On the continuum of care, all studies addressed diagnosis and treatment; none addressed integration of health promotion and protection, disease prevention, or long term and palliative care.

*Study outcomes.* Of the health system outcomes, predefined using the WHO building blocks, 25 (45%) reported on health service delivery, 13 (23%) on health workforce, 4 (7%) on health information systems, 2 (4%) on health financing, and 3 (5%) on leadership and governance. In terms of outcomes, 52 (93%) reported on health outcomes and 4 (7%) on patient-reported outcomes. Fig 3 shows reported outcomes.

**Health service delivery.** This was the most reported health system outcome. Fifteen articles examined the efficiency of integrated tuberculosis and diabetes care, and reported that the

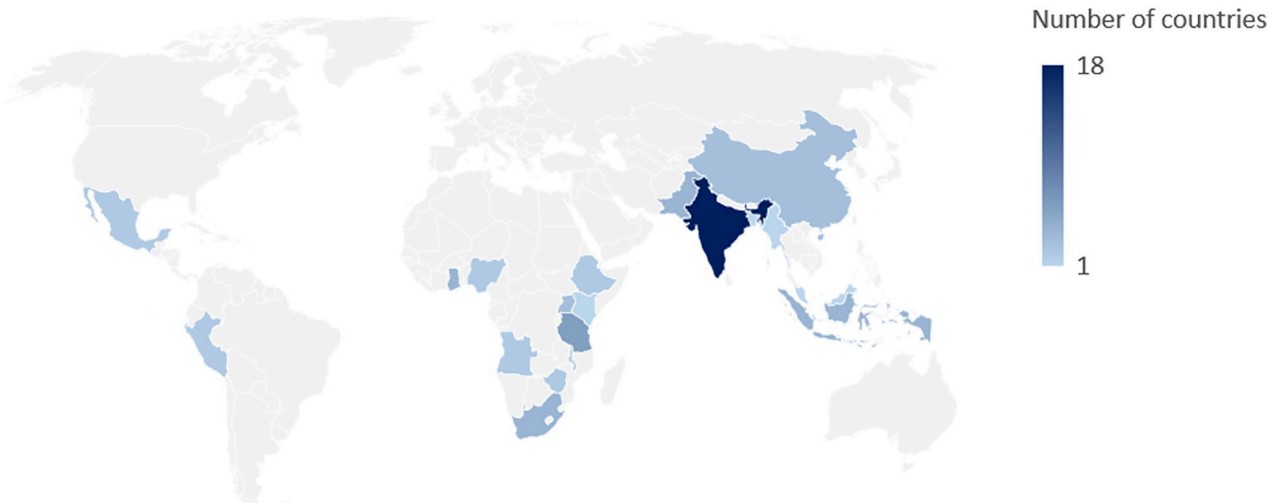

## Map of studies included in systematic review (tuberculosis and diabetes)

**Number of countries**

**Fig 2. Map of studies included in systematic review, by country.**

care process had been simplified [20] and made more efficient by using existing systems and staff [25, 58]. However, inefficiency was also reported in terms of long wait times and increased workload [20, 62]. Two studies reported on adaptability, indicating that the need for, and duration of health worker training reduced over time and their competency with HbA1c testing increased [37, 54]. On adaptability, staff found point of care HbA1c testing to be easy to use to screen for diabetes, although some had mixed feelings about (increased) workload [20, 37].

The health care process and models were reported in ten studies. They examined screening by community health workers and confirmation by clinicians in Pakistan [23] and collaborations between tuberculosis and diabetes clinics in Nigeria [33]. However, lack of equipment and supplies, especially for diabetes, hindered the integration process in India [20]. Salifu et al reported that in Nigeria, the screening was not fully integrated in primary health care or outpatient departments and thus was seen as a supplementary service [62]. In addition, the process of screening for diabetes among tuberculosis patients was not as well defined as tuberculosis screening for diabetes patients and this was noted as an impediment to integrated care. Tuberculosis health workers did not actively screen their patients for diabetes; therefore, any diabetes detection was purely accidental. In a Zimbabwe pilot of 10 facilities services were reorganized to include nurse led screening with referral of difficult cases to the doctor and higher level facilities as necessary [43]. Care coordination was reported in five studies, entailing coordination of multidisciplinary health workers [30, 33, 35], and institutionalization of processes and guidelines for TB and diabetes to facilitate integrated care [62].

Cost-effectiveness is a necessary aspect of health system interventions. There has been a strong presumption that integrated interventions would be cost-effective, relative to separate disease management [13, 126]. Yet, the economic evidence of greater cost-effectiveness is sparse and exclusively related to integrated care for HIV-positive people. We explored outcomes in that literature to identify major research gaps. Aspects of cost were reported in five studies. Lack of supplies and laboratory investigations at hospitals results in out-of-pocket

**Table 2. Baseline characteristics of included studies.**

| Authors | Country | WHO region | World Bank income group | Study design | Target population | Critical Appraisal of evidence | Integration Dyad |
|---|---|---|---|---|---|---|---|
| Achanta et al, 2013 [19] | India | South-east Asia region | Lower middle-income | Cross Sectional-Prevalence study | Tuberculosis patients | High | TB and Diabetes |
| Anand et al, 2018 [20] | India | South-east Asia region | Lower middle-income | Analytical Cross Sectional Study | Tuberculosis patients | Moderate | TB and Diabetes |
| Arini, Sugiyo, and Permana 2022 [21] | Indonesia | South-east Asia | Lower middle-income | Descriptive qualitative | Tuberculosis patients | Moderate | TB and Diabetes |
| Asante-Poku et al, 2019 [22] | Ghana | African region | Lower middle-income | Analytical Cross Sectional | Tuberculosis patents | Moderate | TB and Diabetes |
| Basir et al, 2019 [23] | Pakistan | Mediterranean region | Lower middle-income | Cross Sectional-Prevalence study | Tuberculosis and /or Diabetes patients | Moderate | TB and Diabetes |
| Berkowitz et al, 2018 [24] | South Africa | African region | Upper middle-income | Analytical Cross Sectional | Diabetes patients | High | TB and Diabetes |
| Brey et al, 2020 [25] | South Africa | African region | Upper middle-income | Text and opinion | TB and/or Diabetes -Chronic disease (HIV, TB, Diabetes, Asthma, COPD, hypertension) | High | TB and Diabetes |
| Caceres Calderon, & Ugarte-Gil, 2022 [26] | Global | Global | Global | Review | Tuberculosis patients | Low | TB and Diabetes |
| Chachra et al, 2014 [27] | India | South-east Asia region | Lower middle-income | Cross Sectional-Prevalence study | Tuberculosis patents | Moderate | TB and Diabetes |
| Chamba et al, 2022 [28] | Tanzania | African Region | Lower-middle income | Qualitative study | Tuberculosis patients | Moderate | TB and Diabetes |
| Chamie et al, 2012 [29] | Uganda | African region | Low income | Analytical Cross Sectional | TB and/or Diabetes and other disease (HIV, Malaria, Hypertension) | High | TB and Diabetes |
| Contreras et al, 2017 [30] | Peru | Region of the Americas | Upper middle-income | Cross Sectional-Prevalence study | Tuberculosis patents | Moderate | TB and Diabetes |
| Foo et al, 2022 [31] | Global | Global | Lower middle-income | Systematic review | Tuberculosis patients | Moderate | TB and Diabetes |
| Deepak et al, 2018 [32] | India | South-east Asia region | Lower middle-income | Analytical Cross Sectional | Tuberculosis patents | High | TB and Diabetes |
| Ekeke et al, 2020 [33] | Nigeria | African region | Lower middle-income | Cross Sectional-Prevalence study | Diabetes patients | High | TB and Diabetes |
| Faurholt-Jepsen et al, 2012 [34] | Tanzania | African region | Lower middle-income | Prospective cohort | Tuberculosis patients | High | TB and Diabetes |
| Gnanasan et al, 2011 [35] | Malaysia | Western Pacific region | Upper middle-income | Prospective cohort | Tuberculosis and Diabetes patents | Low | TB and Diabetes |
| Habib et al, 2020 [36] | Pakistan | Mediterranean region | Lower middle-income | Cross Sectional-Diagnostic test accuracy | Tuberculosis and Diabetes patients | Low | TB and Diabetes |
| Huangfu et al, 2019 [37] | Indonesia, Peru, South Africa | South-east Asia region, Region of the Americas, African region | Upper middle-income | Analytical Cross Sectional | Tuberculosis patients | High | TB and Diabetes |
| Jerene et al, 2017 [38] | Ethiopia | African region | Low income | Analytical Cross Sectional | Tuberculosis and/or Diabetes, HIV | High | TB and Diabetes |
| Jiang et al, 2022 [39] | Indonesia | South-east Asia | Lower middle-income | Cross-sectional study | Tuberculosis and/or Diabetes patients | Moderate | TB and Diabetes |

(*Continued*)

**Table 2.** (Continued)

| Authors | Country | WHO region | World Bank income group | Study design | Target population | Critical Appraisal of evidence | Integration Dyad |
|---------|---------|------------|-------------------------|--------------|-------------------|--------------------------------|------------------|
| Joshi et al, 2022 [40] | India | South-east Asia | Lower middle-income | Cluster randomized controlled trial with mixed methods evaluation. | Tuberculosis patients | Moderate | TB and Diabetes |
| Khanna et al, 2013 [41] | India | South-east Asia region | Lower middle-income | Retrospective cohort study | Tuberculosis patients | Moderate | TB and Diabetes |
| Kornfeld et al, 2016 [42] | India | South-east Asia region | Lower middle-income | Prospective cohort | Tuberculosis patients | High | TB and Diabetes |
| Koya et al, 2022 [43] | India | South-east Asia | Lower-middle income | Exploratory in-depth interviews and focus group discussions | Tuberculosis patients | Moderate | TB and Diabetes |
| Kumpatla et al, 2013 [44] | India | South-east Asia region | Lower middle-income | Cross Sectional-Prevalence study | Diabetes patients | Moderate | TB and Diabetes |
| Li et al, 2012 [45] | China | Western Pacific region | Upper middle-income | Analytical Cross Sectional | Tuberculosis patients | Moderate | TB and Diabetes |
| Mishra et al, 2020 [46] | India | South-east Asia region | Lower middle-income | Prospective cohort | Tuberculosis and Diabetes patients | Moderate | TB and Diabetes |
| Mnyambwa et al, 2021 [47] | Tanzania, Kenya, and Uganda | African Region | Lower middle-income, lower middle-income, and low-income (respectively) | Retrospective study | Tuberculosis and/or Diabetes patients | Moderate | TB and Diabetes |
| Mohammed et al, 2021 [48] | Ethiopia | African Region | Low-income | Facility-based study | Tuberculosis patients | Moderate | TB and Diabetes |
| Mukhtar et al, 2017 [49] | Pakistan | Mediterranean region | Lower middle-income | Prospective cohort | Tuberculosis patients | High | TB and Diabetes |
| Mukhtar et al, 2018 [50] | Pakistan | Mediterranean region | Lower middle-income | Prospective cohort | Tuberculosis patients | High | TB and Diabetes |
| Munseri et al, 2019 [51] | Tanzania | African region | Lower middle-income | Analytical Cross Sectional | Tuberculosis patients | High | TB and Diabetes |
| Naik et al, 2013 [52] | India | South-east Asia region | Lower middle-income | Cross Sectional-Prevalence study | Tuberculosis patients | High | TB and Diabetes |
| Nair et al, 2013 [53] | India | South-east Asia region | Lower middle-income | Analytical Cross Sectional | Tuberculosis and/ or Diabetes patients | High | TB and Diabetes |
| Ncube et al, 2019 [54] | Zimbabwe | African region | Lower middle-income | Cross Sectional-Prevalence study | Tuberculosis patients | High | TB and Diabetes |
| Nimkar et al, 2020 [55] | India | South-east Asia region | Lower middle-income | Cross Sectional-Prevalence study | Diabetes patients | Moderate | TB and Diabetes |
| Nyirenda et al, 2022 [56] | Malawi | African Region | Low-income | Retrospective chart review analysis | Tuberculosis patients | Moderate | TB and Diabetes |
| Nyirenda et al, 2023 [57] | Zimbabwe, Angola, Mexico, India, Uganda, Indonesia, and China | Global | Lower middle-income | Qualitative study | Tuberculosis and/or Diabetes patients | Moderate | TB and Diabetes |
| Prakash et al, 2013 [58] | India | South-east Asia region | Lower middle-income | Cross Sectional-Prevalence study | Tuberculosis and/ or Diabetes patients | High | TB and Diabetes |
| Rekha et al, 2007 [59] | India | South-east Asia region | Lower middle-income | Retrospective cohort | Tuberculosis and/ or Diabetes patients | High | TB and Diabetes |

(*Continued*)

**Table 2.** (Continued)

| Authors | Country | WHO region | World Bank income group | Study design | Target population | Critical Appraisal of evidence | Integration Dyad |
|---|---|---|---|---|---|---|---|
| Restrepo et al, 2011 [60] | Mexico | Region of the Americas | Upper middle-income | Analytical Cross Sectional | Tuberculosis patients | Moderate | TB and Diabetes |
| Rohwer et al, 2021 [61] | Global | Global | Global | Systematic review | Tuberculosis patients | High | TB and Diabetes |
| Salifu & Hlongwana, 2021 [62] | Ghana | African Region | Lower-middle income | Exploratory qualitative study | Tuberculosis patients | Moderate | TB and Diabetes |
| Salifu & Hlongwana, 2021 [63] | Ghana | African Region | Lower-middle income | Grounded theory design study | Tuberculosis patients | Moderate | TB and Diabetes |
| Salifu & Holongwa 2021 [64] | Global | Global | Upper middle-income, lower middle-income, and low-income | Scoping review | Tuberculosis patients | Moderate | TB and Diabetes |
| Sarker et al, 2016 [65] | Bangladesh | South-east Asia region | Lower middle-income | Analytical Cross Sectional | Tuberculosis patients | Moderate | TB and Diabetes |
| Sarvamangala et al, 2014 [66] | India | South-east Asia region | Lower middle-income | Cross Sectional-Prevalence study | Tuberculosis patients | Low | TB and Diabetes |
| Segafredo et al, 2019 [67] | Angola | African region | Lower middle-income | Analytical Cross Sectional | Tuberculosis patients | High | TB and Diabetes |
| Shayo et al, 2019 [68] | Tanzania | African region | Lower middle-income | Cross Sectional-Prevalence study | Diabetes clinics | High | TB and Diabetes |
| Shayo & Shayo, 2021 [69] | Tanzania | African Region | Lower-middle income | Secondary data analysis of a cross-sectional survey | Tuberculosis patients | Moderate | TB and Diabetes |
| Sinha et al, 2018 [70] | South Africa | African region | Upper middle-income | Analytical Cross Sectional | Tuberculosis and/ or Diabetes patients | Moderate | TB and Diabetes |
| Ugoeze et al, 2020 [71] | Nigeria | African region | Lower middle-income | Analytical Cross Sectional | Tuberculosis patients | Moderate | TB and Diabetes |
| Xiao et al, 2021 [72] | China | Western Pacific Region | Upper middle-income | Retrospective study | Tuberculosis and/or Diabetes patients | Moderate | TB and Diabetes |
| Zayar et al, 2022 [73] | Myanmar | South-east Asia | Lower-middle income | Cross-sectional study | Tuberculosis patients | High | TB and Diabetes |
| Zhang et al, 2015 [74] | India | South-east Asia region | Lower middle-income | Analytical Cross Sectional | Diabetes, Tuberculosis suspects and contacts | Moderate | TB and Diabetes |
| Abete et al, 2021 [75] | Global | Global | Global | Text and opinion | Diabetes | Moderate | COVID-19, Diabetes, Obesity, and CVD |
| Anjana et al, 2020 [76] | India | South-east Asia region | Lower-middle income | Analytical cross sectional | Type 2 diabetes | Moderate | COVID-19, Diabetes, Obesity, and CVD |
| Ascencio-Montiel et al, 2022 [77] | Mexico | Regions of the Americas | Upper middle-income | Cross sectional | Cardiovascular disease | Moderate | COVID-19, Diabetes, Obesity, and CVD |
| Atef, Gaber, & Zarif, 2022 [78] | Egypt | Eastern Mediterranean Region | Lower-middle income | Text and opinion | Cardiovascular disease | Moderate | COVID-19, Diabetes, Obesity, and CVD |
| Brey et al, 2020 [25] | South Africa | African region | Upper-middle income | Text and opinion | Diabetes, Hypertension | High | COVID-19, Diabetes, Obesity, and CVD |

**Table 2.** (Continued)

| Authors | Country | WHO region | World Bank income group | Study design | Target population | Critical Appraisal of evidence | Integration Dyad |
|---|---|---|---|---|---|---|---|
| Calvert et al, 2022 [79] | South Africa | African Region | Upper middle-income | Analytical cross sectional | Cardiovascular disease | Moderate | COVID-19, Diabetes, Obesity, and CVD |
| Catic et al, 2020 [80] | Bosnia and Herzegovina | European region | Upper-middle income | Analytical cross sectional | Diabetes | Moderate | COVID-19, Diabetes, Obesity, and CVD |
| Chen & Cheng 2022 [81] | Global | Global | Lower-middle income | Text and opinion | Diabetes | Moderate | COVID-19, Diabetes, Obesity, and CVD |
| Cheng et al, 2020 [82] | China | Western Pacific region | Upper-middle income | Text and opinion | Cardiovascular disease | High | COVID-19, Diabetes, Obesity, and CVD |
| Co et al, 2020 [83] | Philippines | Western Pacific region | Lower-middle income | Text and opinion | Stroke | High | COVID-19, Diabetes, Obesity, and CVD |
| Concepcin Zavaleta et al, 2020 [84] | Peru | Region of the Americas | Upper-middle income | Descriptive case report | Type 2 diabetes | Moderate | COVID-19, Diabetes, Obesity, and CVD |
| Da Silva Aquino et al, 2022 [85] | Brazil | Regions of the Americas | Upper middle-income | Quasi-experiment study | Cardiovascular disease | Moderate | COVID-19, Diabetes, Obesity, and CVD |
| David et al, 2022 [86] | South Africa | African Region | Upper middle-income | Qualitative study | Diabetes | Moderate | COVID-19, Diabetes, Obesity, and CVD |
| Di Tommaso, 2020 [87] | Argentina | Region of the Americas | Upper-middle income | Analytical cross sectional | Cardiovascular disease | Moderate | COVID-19, Diabetes, Obesity, and CVD |
| Ding et al, 2020 [88] | China | Western Pacific region | Upper-middle income | Analytical cross sectional | Diabetes, Cardiovascular disease | Moderate | COVID-19, Diabetes, Obesity, and CVD |
| Farooqi et al, 2021 [89] | Pakistan | Eastern Mediterranean Region | Lower-middle income | Text and opinion | Diabetes, Cardiovascular disease | Moderate | COVID-19, Diabetes, Obesity, and CVD |
| Gaspar et al, 2021 [90] | Brazil | Regions of the Americas | Upper middle-income | Analytical cross sectional | Cardiovascular disease | Moderate | COVID-19, Diabetes, Obesity, and CVD |
| Ghosh et al, 2020 [91] | India | South-east Asia region | Lower-middle income | Analytical cross sectional | Type 2 diabetes | Moderate | COVID-19, Diabetes, Obesity, and CVD |
| Girija et al, 2022 [92] | India | South-east Asia | Lower-middle income | Retrospective case series | Cardiovascular disease | Moderate | COVID-19, Diabetes, Obesity, and CVD |

(*Continued*)

**Table 2.** (Continued)

| Authors | Country | WHO region | World Bank income group | Study design | Target population | Critical Appraisal of evidence | Integration Dyad |
|---|---|---|---|---|---|---|---|
| Gona et al, 2020 [93] | India | South-east Asia region | Lower-middle income | Analytical cross sectional | Cardiovascular disease | Moderate | COVID-19, Diabetes, Obesity, and CVD |
| Harindhanavudhi et al, 2022 [94] | Thailand | South-east Asia | Upper middle-income | Analytical cross sectional | Diabetes | Moderate | COVID-19, Diabetes, Obesity, and CVD |
| Joshi et al, 2020 [95] | India | South-east Asia region | Lower-middle income | Analytical cross sectional | Type 2 diabetes | Moderate | COVID-19, Diabetes, Obesity, and CVD |
| Kamvura et al, 2021 [96] | Zimbabwe | African Region | Lower-middle income | Qualitative study | Diabetes, Cardiovascular disease | Moderate | COVID-19, Diabetes, Obesity, and CVD |
| Kolesnyk et al, 2021 [97] | Ukraine | European Region | Lower-middle income | Focus group discussions | Diabetes, Cardiovascular disease | Moderate | COVID-19, Diabetes, Obesity, and CVD |
| Krisiunas et al, 2020 [98] | Rwanda | African region | Low income | Text and opinion | Type 2 diabetes | High | COVID-19, Diabetes, Obesity, and CVD |
| Kyazze et al, 2021 [99] | Africa | African Region | Lower-middle income | Literature review | Diabetes | Moderate | COVID-19, Diabetes, Obesity, and CVD |
| León-Vargas, 2021 [100] | Colombia | Region of the Americas | Upper middle-income | Quasi-experimental study | Diabetes | Moderate | COVID-19, Diabetes, Obesity, and CVD |
| Li et al, 2020 [101] | China | Western Pacific region | Upper-middle income | Analytical cross sectional | Cardiovascular disease | Moderate | COVID-19, Diabetes, Obesity, and CVD |
| Liu et al, 2020 [102] | China | Western Pacific region | Upper-middle income | Text and opinion | Diabetes | High | COVID-19, Diabetes, Obesity, and CVD |
| Mishra et al, 2021 [46] | India | South-east Asia | Lower middle-income | Case report | Diabetes, Cardiovascular disease | High | COVID-19, Diabetes, Obesity, and CVD |
| Mistry et al, 2021 [103] | Bangladesh | South-east Asia | Lower-middle income | Analytical cross sectional | Diabetes, Cardiovascular disease | Moderate | COVID-19, Diabetes, Obesity, and CVD |
| Mohan et al, 2022 [104] | India | South-east Asia | Lower middle-income | Quasi-experimental study | Cardiovascular disease | Moderate | COVID-19, Diabetes, Obesity, and CVD |
| Nan et al, 2020 [105] | China | Western Pacific region | Upper-middle income | Quasi-experimental study | Cardiovascular disease | Moderate | COVID-19, Diabetes, Obesity, and CVD |

*(Continued)*

**Table 2.** (Continued)

| Authors | Country | WHO region | World Bank income group | Study design | Target population | Critical Appraisal of evidence | Integration Dyad |
|---|---|---|---|---|---|---|---|
| Nanditha et al, 2021 [106] | India | South-east Asia | Lower middle-income | Text and opinion | Diabetes | High | COVID-19, Diabetes, Obesity, and CVD |
| Okpara & Oghagbon, 2021 [107] | Africa | African Region | Lower-middle income | Text and opinion | Cardiovascular disease | Moderate | COVID-19, Diabetes, Obesity, and CVD |
| Olickal et al, 2020 [108] | India | South-east Asia region | Lower-middle income | Analytical cross sectional | Diabetes | Moderate | COVID-19, Diabetes, Obesity, and CVD |
| Owopetu et al, 2021 [109] | Sub-Saharan Africa | African Region | Lower-middle income | Text and opinion | Diabetes, Cardiovascular disease | Moderate | COVID-19, Diabetes, Obesity, and CVD |
| Pandian et al, 2021 [110] | Asia | South-east Asia | Lower-middle income | Text and opinion | Cardiovascular disease | High | COVID-19, Diabetes, Obesity, and CVD |
| Queiroz et al, 2020 [111] | Brazil | Region of the Americas | Upper-middle income | Analytical cross sectional | Type 2 diabetes | Moderate | COVID-19, Diabetes, Obesity, and CVD |
| Ratnayake et al, 2022 [112] | Jordan | Eastern Mediterranean Region | Upper middle-income | Cohort study | Cardiovascular disease | Moderate | COVID-19, Diabetes, Obesity, and CVD |
| Sodipo et al, 2021 [113] | Nigeria | African Region | Lower middle-income | Analytical cross sectional | Diabetes | Moderate | COVID-19, Diabetes, Obesity, and CVD |
| Tong et al, 2022 [114] | China | Western Pacific Region | Upper middle-income | Analytical cross sectional | Cardiovascular disease | Moderate | COVID-19, Diabetes, Obesity, and CVD |
| Tran et al, 2021 [115] | Kenya | African Region | Lower-middle income | Text and opinion | Diabetes, Cardiovascular disease | Moderate | COVID-19, Diabetes, Obesity, and CVD |
| Wang et al, 2021 [116] | China | Western Pacific Region | Upper middle-income | Quasi-experiment study | Diabetes, Cardiovascular disease | High | COVID-19, Diabetes, Obesity, and CVD |
| Zafra-Tanaka et al, 2022 [117] | Peru | African Region | Upper middle-income | Qualitative study | Stroke | Moderate | COVID-19, Diabetes, Obesity, and CVD |
| Zhao et al, 2020 [118] | Global | Global | Global | Analytical cross sectional | Stroke | Moderate | COVID-19, Diabetes, Obesity, and CVD |
| Arini et al, 2022 [21] | Indonesia | South-east Asia region | Lower middle-income | Qualitative study | Diabetes, Tuberculosis | Moderate | COVID-19, Tuberculosis, and Diabetes |

(*Continued*)

**Table 2.** (Continued)

| Authors | Country | WHO region | World Bank income group | Study design | Target population | Critical Appraisal of evidence | Integration Dyad |
|---|---|---|---|---|---|---|---|
| Brault et al, 2021 [119] | Sub-Saharan Africa | African region | Multi-income | Text and Opinion | Diabetes, Tuberculosis | High | COVID-19, Tuberculosis, and Diabetes |
| Caceres, Calederon, Ugarte, 2022 [26] | Global | Global | Multi-income | Text and Opinion | Diabetes, Tuberculosis | Low | COVID-19, Tuberculosis, and Diabetes |
| Kavenga et al., 2021 [120] | Zimbabwe | African region | Low income | Analytical Cross sectional | Diabetes, Tuberculosis | Moderate | COVID-19, Tuberculosis, and Diabetes |
| Loveday et al, 2020 [121] | Global | Global | Multi-income | Text and Opinion | Diabetes, Tuberculosis | Moderate | COVID-19, Tuberculosis, and Diabetes |
| Nesan et al, 2021 [122] | India | South-east Asia region | Lower middle-income | Quasi-experimental study | Diabetes, Tuberculosis | Moderate | COVID-19, Tuberculosis, and Diabetes |
| Njau et al, 2022 [123] | Kenya | African region | Lower middle-income | Analytical Cross sectional | Diabetes, Tuberculosis | Moderate | COVID-19, Tuberculosis, and Diabetes |
| Visca et al, 2021 [124] | Global; Sierra Leone | Global | Low income | Text and Opinion | Diabetes, Tuberculosis | Moderate | COVID-19, Tuberculosis, and Diabetes |
| Williams et al, 2022 [125] | Eswatini | African region | Lower middle-income | Qualitative study | Diabetes, Tuberculosis | Moderate | COVID-19, Tuberculosis, and Diabetes |

expenditures for laboratory tests such as random or fasting blood glucose, HbA1c for diabetes, radiological procedures e.g., chest x-rays or clinical procedures such as spirometry for tuberculosis [20, 23, 25, 29, 62]. A few studies reported on the incremental costs of integrating services. A best-case example of systems integration was medicine home delivery during the COVID-19 pandemic in South Africa, that resulted in no incremental costs except transportation costs, as they had used health systems that were already financed [25]. Chamie et al reported that the incremental cost of adding diabetes and hypertension screening to HIV services was US $2.41, compared to $4.58 for adding TB (rapid and PCR testing) [29].

**Health workforce** was reported on by thirteen studies. In eight studies joint training occurred either onsite or via training workshops with training subsequently cascaded to other health workers [20, 29, 38, 45]. Health workers felt that they lacked the skills and training to deliver integrated care [20]. Integrated care involved multidisciplinary teams including doctors, nurses, psychologists, and social workers among others [30]. Task shifting and sharing was also used as an approach to deliver integrated care in four studies. A best case example was in Ghana where screening tasks were shifted to a specific staff member–the TB task shifting officer, resulting in successful service integration [54]. Task sharing entailed screening and referral by community health workers, uncomplicated case management by nurses, referral to doctors for complicated cases, and working with pharmacists to develop treatment plans for tuberculosis and diabetes [25, 35, 62]. A good team-based care example was seen in Malaysia where a pharmacist-led service for patients with tuberculosis and diabetes identified medication related problems, aided goal setting, developed treatment plans, and undertook monitoring and follow up. They also made recommendations to physicians regarding identified problems [35].

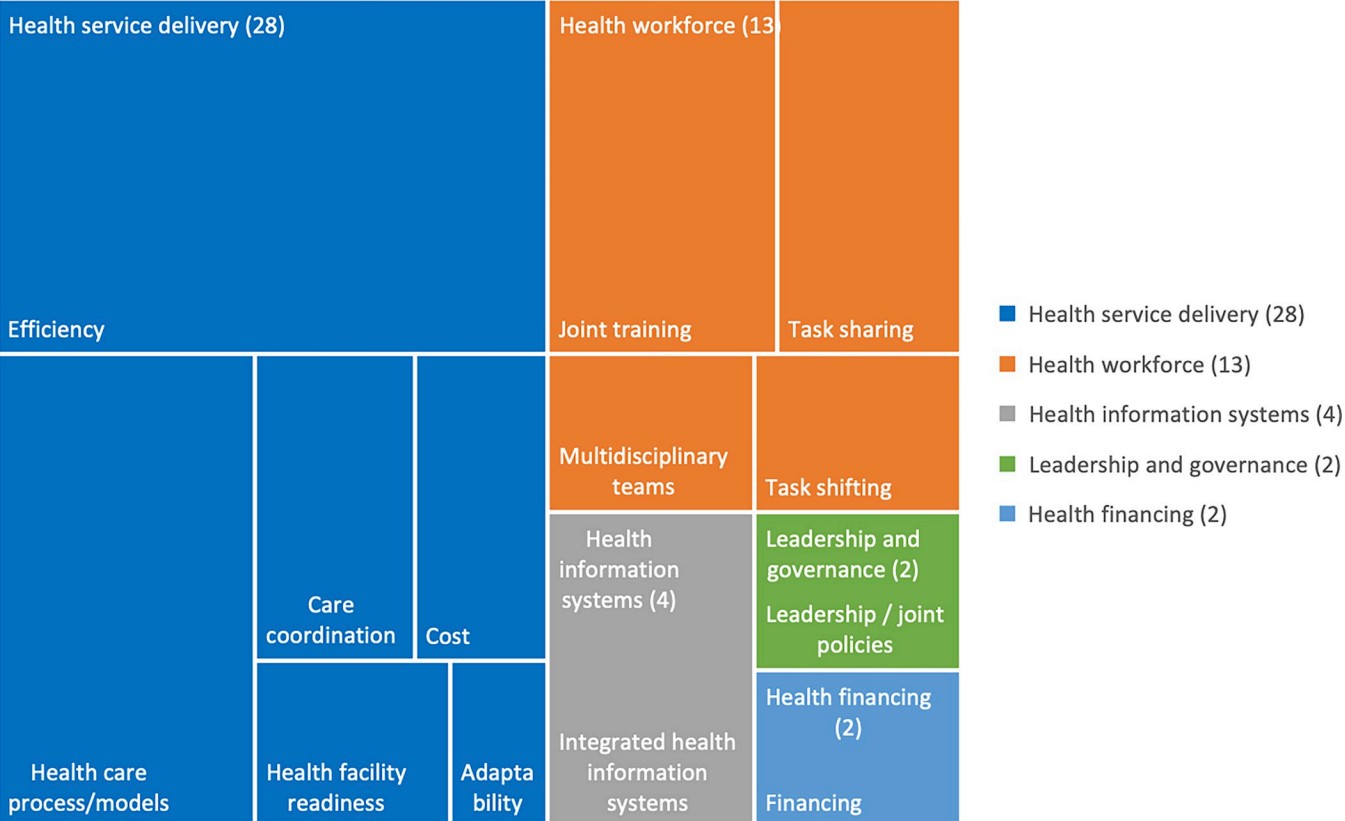

**Fig 3. Tree map of health system outcomes identified in the literature (as per the WHO building blocks).**

**Health information systems.** Only four studies reported on integrated health information systems. One example is use of pre-existing data to address integration [70]. A lack of standardized recording and reporting tools for integrated services, especially diabetes, was noted [20]. Some studies reported development of specific tuberculosis-diabetes registers for this purpose [41, 70].

**Essential medicines.** No studies were identified that reported on integrated systems for joint/pooled procurement or integrated supply chain and logistics management systems for essential medicines.

**Health financing.** Two studies noted disparities in funding for bidirectional screening. Specifically, integration of diabetes screening for tuberculosis patients was funded through the national tuberculosis program, whereas tuberculosis screening was mainly funded out-of-pocket by diabetes patients [62].

**Leadership and governance.** Two studies reported on the leadership and governance of tuberculosis and diabetes integration. Inclusion of screening guidelines and recommendations in clinic policies and documents and receiving support from the clinics or departments were seen as enablers of integrated care [20, 62]. However, it was also noted that tuberculosis and diabetes units are managed under different administrative divisions, making it harder to integrate services.

*Barriers and facilitators of integration of tuberculosis and diabetes.* Barriers to integrated care included delays in screening, fear and stigmatization of tuberculosis, poor collaboration between TB and DM units, skewed funding for screening, long waiting times, costs involved

with blood tests, increased workload for health workers, inadequate medical supplies, inadequate skills and knowledge of health workers, lack of standardized reporting tools for diabetes, and incorrect contact information that hinders contacting patients. Key facilitators included increased staff capacity, institutionalization of bidirectional screening through guidelines and staff, and availability of standardized screening tools for tuberculosis.

## COVID-19 and diabetes, obesity, and CVD integration

**Search results.** The search identified a total of 1,797 records that underwent title and abstract screening to remove duplicates. Of these, 122 articles were included in the full text assessment, resulting in 46 articles included in the review.

The 46 included articles were conducted in 20 countries, representing all WHO global regions, with the greatest number conducted in India (n = 9) and China (n = 7). Studies represented low, lower-middle, and upper-middle income countries, with most in upper-middle income countries (n = 21). Three articles collected data from multiple countries. In terms of disease area, 26 articles focused on diabetes (11 on type 2 diabetes, 1 on type 1 diabetes, and 14 on diabetes of unspecified type), and 25 on cardiovascular diseases (5 specifically on stroke). No articles examined obesity and COVID-19. Table 2 summarizes included articles on COVID-19 and diabetes, obesity, and cardiovascular disease.

Most studies were cross sectional (n = 29), or short opinion pieces (n = 19) that shared best practices and learnings for responding to the COVID-19 pandemic. As a result, learnings regarding the introduction of new approaches were largely descriptive, rather than the result of rigorous evaluation. All cross-sectional studies and the case series report were determined as moderate quality, and the "text and opinion" studies as high quality.

We found little direct experience to guide structural approaches to integrating cardiometabolic care with non-emergency ID treatment, yet some of the pandemic responses may be adapted to ongoing care needs. The interventions were population level health service adaptations to respond to the pandemic by reducing COVID-19 exposure risk and maintaining routine care access during clinic closures and lockdowns. Of the 46 included studies, the focus was on COVID and diabetes integration (n = 17), cardiovascular diseases (n = 16), stroke (n = 3), and combinations of these conditions (n = 10).

Most articles (n = 33) explored telemedicine and digital solution introduction for patient-provider communication, such as consultations via WhatsApp or video calling platforms. For services that remained in person, several articles (n = 8) shared guidelines and lessons learned for revising operating procedures or treatment protocols to reduce COVID-19 exposure risk among health providers. Finally, seven studies examined approaches for medicine home delivery for patients when lockdowns constrained their ability to collect prescriptions. All of the above innovations are being explored as potential health system improvements to improve accessibility and responsiveness to patient needs beyond pandemic conditions.

Many adaptations were secondary (n = 14), meaning that they were instituted after the onset of the COVID-19 pandemic and were intended to reduce escalation. Twelve articles described tertiary adaptations—to minimize adverse events associated with COVID-19. Six articles examined primary adaptation examples, i.e. adaptions made before the COVID-19 onset, such as changing appointment schedules or diabetes or CVD patient workflow. As most studies examined the introduction of telemedicine services, the most common adaptive capacity was available technology (n = 33). Others included: available information and skills (n = 6), infrastructure (n = 3), and resources (n = 1), all of which related to new care delivery approaches and adjustments to routine procedures in response to the pandemic. For example, the Chinese Society of Cardiology issued an expert consensus on cardiovascular disease

**Table 3. Summary of COVID-19 and diabetes, obesity and CVD articles by type and scope of integration.**

| Integration: Type and Scope | N (%) |
|---|---|
| **Intervention** | |
| Telemedicine (n = 33) | 33 (71.7) |
| Revisions in care guidelines (n = 6) | 6 (13.0) |
| Medicine delivery (n = 7) | 7 (15.2) |
| **Type of adaptation** | |
| Primary | 6 (13.0) |
| Secondary | 14 (30.4) |
| Tertiary | 12 (26.1) |
| **Adaptive capacity** | |
| Availability of resources | 1 (2.2) |
| Available technology | 33 (71.7) |
| Available information and skills | 6 (13.0) |
| Infrastructure | 3 (6.5) |
| Institutions | 0 (0.0) |
| Equity | 0 (0.0) |
| **Focus** | |
| Population-based | 26 (56.5) |
| Person-based | 20 (43.5) |

management during COVID-19, which improved information and skills training to enable clinicians to alter treatment protocols for cardiovascular emergencies and reduce COVID-19 exposure risk [82]. Table 3 summarizes included articles by integration type and scope.

The aim of most studies was to assess the feasibility of health service delivery adaptations in response to the COVID-19 pandemic (Fig 4). As a result, 41 studies reported health service

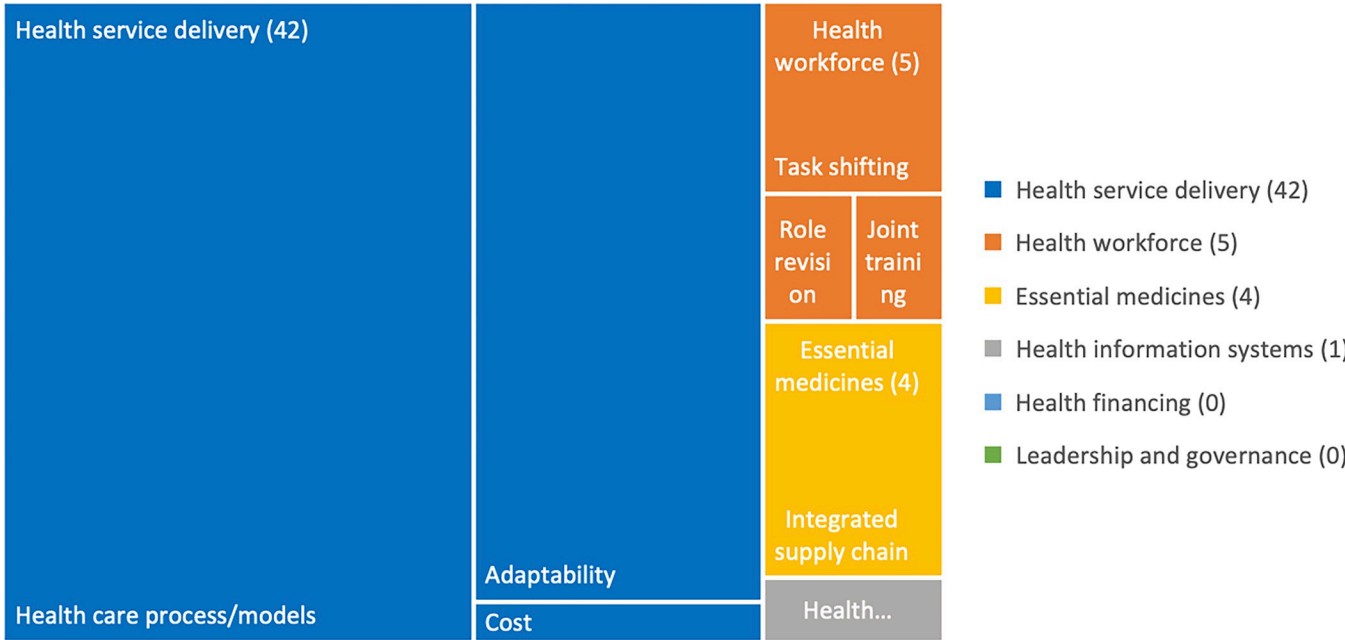

**Fig 4. Tree map of COVID-19 and diabetes, obesity and CVD articles by study outcomes.**

delivery outcomes, namely the healthcare process and models (n = 15) [40, 80, 83, 87, 88, 91, 93, 95, 98, 101, 102, 105, 108, 111, 118], adaptability (n = 15) [76, 82, 83], efficiency (n = 11), and cost (n = 1) [25]. The fifteen articles that reported on healthcare processes and models shared the design and feasibility of new models of care to sustain services during pandemic disruptions. New models of care included telemedicine consultations for routine diabetes and cardiovascular patient care, an internet-based treatment algorithm for diabetic foot ulcers in China [102], and handheld smartphone camera screening for diabetic retinopathy in Brazil [111]. These alternative models were reported to be feasible and acceptable to patients. Reported trade-offs included a lack of in-person connection and inability to conduct a physical examination. Three studies also shared revised treatment practices that their facilities had implemented to reduce COVID-19 exposure risk, such as care for patients with diabetic ketoacidosis using subcutaneous insulin every 4 hours, rather than via an insulin infusion pump, to reduce bedside time, risk of staff exposure, and requirement for personal protective equipment. Finally, one article discussed the establishment of a new 'internet hospital' in China during COVID-19, that provided medicine home delivery [88].

Three studies reported how health service adaptations influenced delivery. These hospital studies in China, India, and the Philippines reported on new procedures to maintain continuity of care while reducing staff COVID-19 exposure risk. Procedures included telemedicine consultation, and home visits for biospecimen collection, which were reported as feasible and acceptable to patients [76]. In China and the Philippines, hospital staff developed new protocols to screen patients for COVID-19 before ward transfer, and coordinated across specialties to improve efficiency and reduce exposure time during diagnostics and other procedures [82].

Thirteen studies assessed the cost of health service delivery adaptations. In South Africa community health workers were engaged to deliver medicines for diabetes, hypertension, and other conditions to patients in their catchment area [25]. By including this service within an established program, the additional costs were reported to be minimal.

Four studies examined health workforce outcomes related to task shifting (n = 3) and role revision (n = 1) to encompass new telemedicine services staffed by clinical pharmacists or medicine home delivery by community health workers [22, 64, 65, 71]. One reported that during the pandemic cardiologists in China established new virtual platforms to coordinate care with primary care doctors [82]. Two studies reported programs in South Africa [25] and China [88] that successfully offered home delivery and/or online prescription fulfilment for medicines for conditions including NCDs. No studies assessed integrated health financing or leadership and governance.

Whereas most studies focused on the design and feasibility of the health system adaptations, nineteen also reported on the associated health and patient-reported outcomes. A study in India reported that participation in a diabetes telemedicine consultation was associated with a significant decrease in weight, body mass index, systolic blood pressure, HbA1c, and serum cholesterol, but significantly increased diastolic blood pressure [76]. Following a telephone consultation intervention led by clinical pharmacists in India, patients reported good adherence to dietary guidance and medication, but lower adherence to physical exercise and glucose monitoring guidance [95]. A study assessing the effectiveness of an app to improve patient-clinician communication and reduce time to access services for ST-Segment Elevation Myocardial Infarction, found time to care was greater after pandemic onset, but there was no difference in short term adverse clinical outcomes between patients who did and did not use the app, including for mortality [105]. Three studies assessed patient satisfaction with telemedicine services [76], and a majority of patients reported satisfaction with their telemedicine experience and interest in future use of such services. Finally, four studies commented on the

value and sustainability of system adaptations to ensure care was provided in the right place at the right time [25, 101, 102, 111].

These findings suggest several barriers and facilitators to adapting health services to include telemedicine services for diabetes and cardiovascular diseases. These services were made feasible by widespread availability of devices that enable telemedicine consultations, such as mobile phones and computers, leadership from expert organizations, and strong coordination and clear communication across specialties. Barriers and limitations included inability to undertake a physical examination, lack of face-to-face connection, and technological difficulties, such as poor or dropped connections. Nonetheless, the extraordinary circumstances engendered by COVID 19 encouraged innovation that reflects the potential for integrated care for chronic and emergent diseases with multiple benefits for patients and health systems. Not wishing to let "a disaster be wasted," the lesson for policymakers is to build more sustained integrated systems and assess the suitability for different comorbid disease combinations.

### COVID-19, tuberculosis, and diabetes integration

**Search results.**    The search identified a total of 84 records that underwent title and abstract screening to remove duplicates. Of these, 15 articles were included in the full text assessment, resulting in 9 articles for inclusion in the review (Fig 5).

## Discussion

The limited literature found by this review across a range of major infectious disease and noncommunicable disease (NCD) dyads suggests that, prior to the COVID-19 pandemic, health

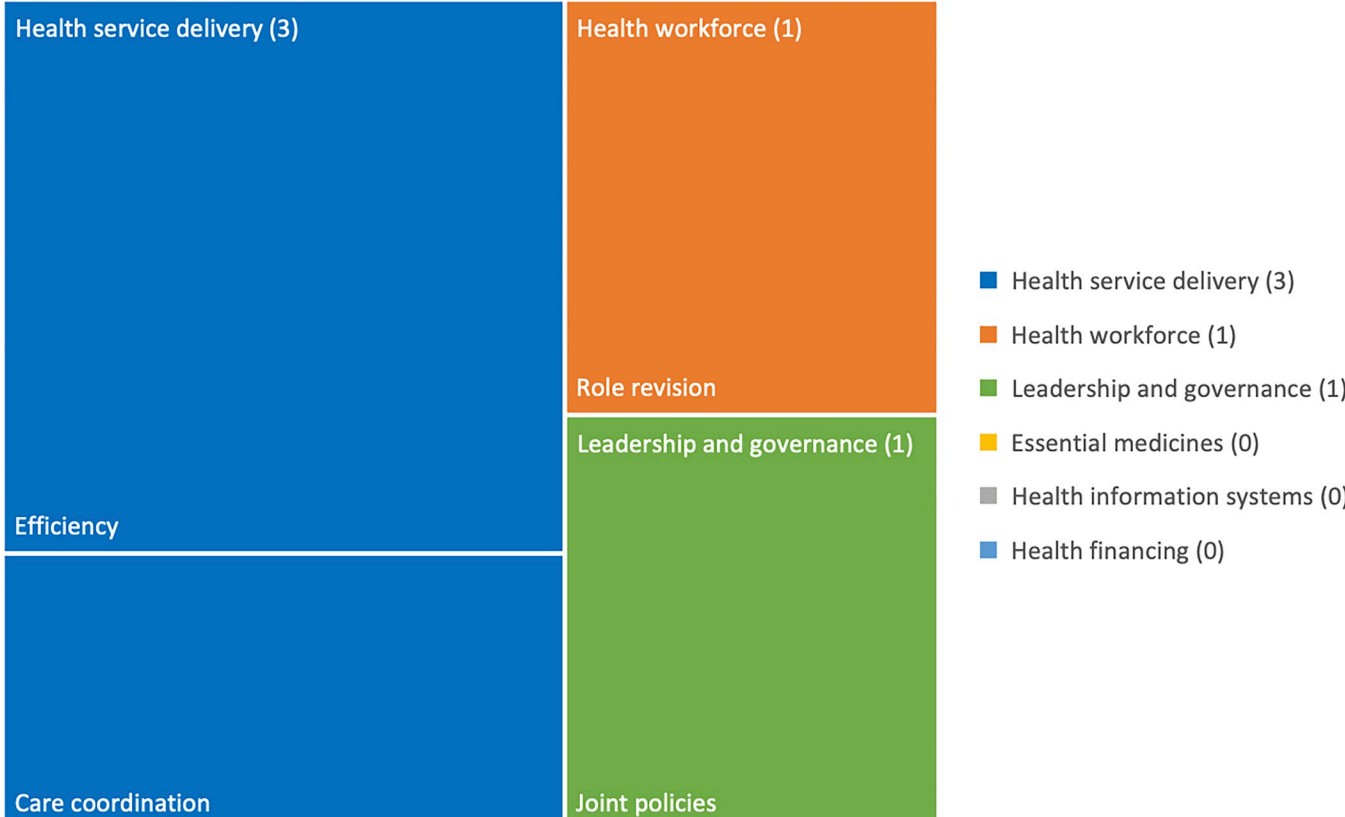

**Fig 5. Tree map of COVID-19, tuberculosis, and diabetes integration articles by study outcomes.**

system ID-NCD integration beyond HIV was substantially limited in low- and middle- income countries. Evidence on NCD—trachoma / onchocerciasis integration was wholly absent, and tuberculosis and diabetes integration efforts predominated. For tuberculosis and diabetes integration there were few studies in low-income countries (LICs) and limited literature in middle income countries (MICs)–only 56 across the search period. This suggests that countries with greater resources have been relatively more proactive to integrate care for these conditions, whereas early LIC developments would appear to have focused on HIV (for which many examples exist), due to greater disease prevalence, resources, and–crucially—donor support to foster integration [9–13, 127, 128]. The onset of COVID-19 produced a surge of integration of COVID-19 and cardiometabolic disease, and COVID-19 and diabetes or tuberculosis, comprising almost half of all studies reviewed.

Nonetheless, a continuing and significant limitation of the identified examples of integration for these conditions was their scope. Examples focused almost exclusively on service delivery, suggesting this has been prioritized rather than integration across health system building blocks, such as integrated workforce capacities or systems, financing, or essential medicines provision. This applied to all ID-NCD dyads in the study. Regarding tuberculosis and diabetes, the scope was even narrower, as service delivery integration almost wholly concentrated upon screening. Few examples addressed treatment beyond initial screening, with care thus continuing to be separate for the conditions. Diagnosis may currently be the most affordable population level intervention when resources are constrained, and developing integrated treatment and long-term care will require longer term investment in building workforce and health information system capacities. A service delivery focus is apparent also in the more plentiful examples of countries' HIV-NCD efforts, yet these commonly extend far more extensively into care delivery, and often encompass preventive services also [9, 11]

The narrow focus upon diabetes and tuberculosis screening also indicates selective implementation of integration guidance. Tuberculosis and diabetes integration studies were mostly published after the Tuberculosis Diabetes collaborative framework in 2012 [129], which may have catalyzed practical examples. Yet whereas the guidance encourages system level integration, such as monitoring and evaluation, most studies reported solely on bidirectional screening. Notably also, there was little or no focus in the published examples on either health promotion or long term and palliative care, meaning that important aspects of the care continuum remain fragmented between infectious and noncommunicable conditions. The review found scant evidence of diabetes and tuberculosis integration across the six health system building blocks, despite global recommendations for a system-wide approach [129]. Possible explanations may include resource constraints, competing priorities, and a lack of leadership. One further reason may be the apparent evidence gap on how to integrate the wider health system building blocks, such as medicines provision, health financing, and leadership and governance, but moreover in relation to outcomes for health systems (and a rationale to prioritize such integration).

Many NCD service delivery programs introduced measures related to the infectious disease COVID-19 during the pandemic. This greatly increased the number of published examples of programs taking an integrated approach to managing one or more NCDs and an infectious disease. The scope of integration followed the same pattern as for tuberculosis or diabetes, with health service delivery integration predominant (42/48 studies)–influenced by the high policy priority upon maintaining essential NCD services. Measures focused on reducing COVID-19 exposure risk and maintaining routine care access for NCD patients. In this way the pandemic—in many cases for the first time—encouraged health service staff in at least a limited way to consider IDs and NCDs together, and in many cases this integrated consideration has persisted, for example, in the continued greater focus given to infection control

within NCD service delivery. The pandemic also produced a substantial research output indicating that untreated NCDs represent vulnerabilities that result in poorer health outcomes, and recommended this be considered within future pandemic planning [130–132]. We propose therefore that COVID-19 may have a legacy in terms of awareness and willingness to consider integrated approaches to address co-occurring health challenges.

Nonetheless, at present the potential for integrated infectious–noncommunicable disease systems to support health systems to manage the growing burden of comorbidity within LMIC populations remains largely unrealized for many conditions of high and growing prevalence. A step change in support by donors and governments to enable health programs to develop, test, and evaluate integration across health system building blocks is needed to generate evidence to catalyze guide policy and practice.

## Limitations

This study focused on English language articles and may have omitted articles in other languages. The search was limited to LMICs and does not address the potential to learn from and critique NCD integration in high-income countries.

## Conclusions

Integrating infectious and noncommunicable diseases has been proposed as a potential way for overloaded health systems to deal with increasingly complex health needs, particularly the growing burden of infectious and NCD comorbidities in LMIC populations. This recommendation is based on the premise of system-wide integration–integrating not only a part of the system, but developing approaches to integrate financing, workforce capacities, access to medicines, and across the continuum from health promotion through to long term and palliative care. Greater cost-effectiveness has also been posited as a benefit of integration, but the evidence is insufficient to draw conclusions. The review indicates that, beyond HIV, piloting and implementing integration related to ID-NCD disease dyads has been limited in extent, and moreover concentrated on very narrow aspects of service delivery–particularly screening. This is the case for even TB-diabetes, where most non-HIV integration attention has focused. Implementation projects–including regarding wider health system building blocks, and robust outcome and process evaluation are needed for countries to assess and maximize the potential for integrated approaches to manage infectious and noncommunicable diseases.

## Supporting information

**S1 Table. Ranking of evidence of included studies.**
(DOCX)

## Acknowledgments

We express gratitude to Erin Eckert who contributed to the early conceptualization of the study.

## Author Contributions

**Conceptualization:** Angela Jackson-Morris, Sarah Masyuko, Erica L. Kocher, Rachel Nugent.

**Data curation:** Sarah Masyuko, Lillian Morrell, Ishu Kataria, Erica L. Kocher.

**Formal analysis:** Angela Jackson-Morris, Sarah Masyuko, Lillian Morrell, Ishu Kataria, Erica L. Kocher.

**Investigation:** Sarah Masyuko, Erica L. Kocher.

**Supervision:** Angela Jackson-Morris, Rachel Nugent.

**Writing – original draft:** Angela Jackson-Morris, Sarah Masyuko, Ishu Kataria, Erica L. Kocher, Rachel Nugent.

**Writing – review & editing:** Angela Jackson-Morris, Lillian Morrell, Ishu Kataria.

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
