## [Decision Letter · Decision Letter 0]

27 Dec 2023

PGPH-D-23-01447

Tackling syndemics by integrating infectious and noncommunicable diseases in health systems of low- and middle-income countries: A narrative systematic review

Dear Dr. Jackson-Morris,

Thank you for submitting your manuscript to PLOS Global Public Health. After careful consideration, we feel that it has merit but does not fully meet PLOS Global Public Health’s publication criteria as it currently stands. Therefore, we invite you to submit a revised version of the manuscript that addresses the points raised during the review process.

Inclusion of HIV/NCD integration data to enhance the narrative. Please provide justifications for the infectious diseases selected for the study and provide a clearer distinction between the management of endemic and epidemic diseases. The reviewers have suggested to refine conclusions to align with the evidence presented. Also, please discuss how the exclusion of non-English language studies might impact the results.

We look forward to receiving your revised manuscript.

Kind regards,

Giridhara R Babu, MBBS, MPH, PhD

Academic Editor

Journal Requirements:

Additional Editor Comments (if provided):

Inclusion of HIV/NCD integration data to enhance the narrative. Please provide justifications for the infectious diseases selected for the study and provide a clearer distinction between the management of endemic and epidemic diseases. The reviewers have suggested to refine conclusions to align with the evidence presented. Also, please discuss how the exclusion of non-English language studies might impact the results.

Reviewers' comments:

Reviewer's Responses to Questions

**Comments to the Author**

1. Does this manuscript meet PLOS Global Public Health’s publication criteria? Is the manuscript technically sound, and do the data support the conclusions? The manuscript must describe methodologically and ethically rigorous research with conclusions that are appropriately drawn based on the data presented.

Reviewer #1: Yes

Reviewer #2: Yes

2. Has the statistical analysis been performed appropriately and rigorously?

Reviewer #1: N/A

Reviewer #2: N/A

3. Have the authors made all data underlying the findings in their manuscript fully available (please refer to the Data Availability Statement at the start of the manuscript PDF file)?

Reviewer #1: Yes

Reviewer #2: Yes

4. Is the manuscript presented in an intelligible fashion and written in standard English?

Reviewer #1: No

Reviewer #2: Yes

5. Review Comments to the Author

Reviewer #1: The manuscript “Tackling syndemics by integrating infectious and noncommunicable diseases in health systems of low- and middle-income countries: A narrative systematic review” is significant in the context of health system preparedness in LMICS. It seems three reviews are present in a single paper. The authors need to address the following suggestions:

Introduction

The rationale of the review needs to be presented in detail.

Why is it important to integrate?

What are the gaps in existing evidence?

Methods

In the analysis section describe how did you prepare the Tree map of Health System Outcomes identified in the literature (as per the WHO building blocks).

Provide a single search strategy for all domains.

Results

Three PRISMA flow diagrams in a single review are confusing and unnecessary increasing the number of figures and tables. Make a single search, and in PRISMA provide the final how many articles were included:

Total studies included (N=) and in these articles

A. TB and diabetes integration (n=)

B. COVID-19 and Diabetes, Obesity, and CVD Integration (n=)

C. COVID-19, Tuberculosis, and Diabetes Integration (n=)

Instead of three Tables on study characteristics, provide a single table with all the included studies. Combine the following tables into a single table. Create one more column on the domain address (A, B, C, AB, BC, CD, ABC).

Table 2: Baseline characteristics of included studies for tuberculosis and diabetes integration

Table 5: Summary of COVID-19 and Diabetes, Obesity and CVD integration articles by study characteristics

Table 7: Summary of COVID-19, Tuberculosis, and Diabetes integration articles by study characteristics.

Combine the three Maps of the included studies (figures 3, 6 and 9).

Present the figures 4, 7 and 10 (Tree Maps) separately (as it is provided).

Discussion

Discuss how the integration is cost-effective.

Write a paragraph on research gaps and implications of the findings for policy and practices.

Reviewer #2: This is an interesting topic, but I have reservations about several decisions made and the overall framing of the paper.

1. The decision not to integrate and include data on HIV/NCD integration is a major omission, since that is the area with the most literature. As such, comparative/narrative synthesis across NCDs is missing the area of literature with the largest body of experience. This HIV literature is not even engaged with in the discussion.

2. The choice of infectious disease entities chosen is not justified and is idiosyncratic. There is very little literature, at all, on onchocerca and trachoma so the review becomes by default a review of TB alone. There are many other high impact infectious diseases, such as malaria, that could have be included.

3. I question the taxonomic decision to include management of endemic infectious diseases, eg tuberculosis, within the same framework as epidemic infectious diseases (COVID). These are completely different phenomena.

4. Along with point 3, while much of the literature related to TB integration do represent true integrations of service lines the COVID related literature is much more about pandemic response adaptations which are not specific to NCDs but occurred across all areas of all health systems globally. These pandemic response adaptations are a distinct category and do not convincingly inform the management of both NCDs and endemic infectious diseases within primary care or integrated care models.

5. The conclusions on the reach and distribution of ID-NCD integration are over-reaching, given the limited search strategies used.

6. The conclusion that ID-NCD integration surged after COVID-19 is also over-reaching, since TB-NCD integration and similar scenarios for other endemic infectious diseases is a completely different phenomenon from the pandemic response and these two cannot be categorized together.

7. The exclusion of studies in languages other than English is a major limitation, and it is not clearly justified given how simple text translation has become using multiple research tools. Although not clear from the methods, the flowcharts seem to show that the language exclusion was applied at the level of the primary search, so we are not even given information on how many identifiable non-English language studies may be excluded here from the analysis.

6. PLOS authors have the option to publish the peer review history of their article (what does this mean?). If published, this will include your full peer review and any attached files.

**Do you want your identity to be public for this peer review?** For information about this choice, including consent withdrawal, please see our Privacy Policy.

Reviewer #1: **Yes: **Krushna Chandra Sahoo

Reviewer #2: **Yes: **Peter Rohloff

---

## [Decision Letter · Decision Letter 1]

26 Mar 2024

Tackling syndemics by integrating infectious and noncommunicable diseases in health systems of low- and middle-income countries: A narrative systematic review

PGPH-D-23-01447R1

Dear Dr Jackson-Morris,

We are pleased to inform you that your manuscript 'Tackling syndemics by integrating infectious and noncommunicable diseases in health systems of low- and middle-income countries: A narrative systematic review' has been provisionally accepted for publication in PLOS Global Public Health.

Best regards,

Giridhara R Babu, MBBS, MPH, PhD

Academic Editor

Reviewer Comments (if any, and for reference):

Reviewer's Responses to Questions

**Comments to the Author**

1. If the authors have adequately addressed your comments raised in a previous round of review and you feel that this manuscript is now acceptable for publication, you may indicate that here to bypass the “Comments to the Author” section, enter your conflict of interest statement in the “Confidential to Editor” section, and submit your "Accept" recommendation.

Reviewer #3: All comments have been addressed

Reviewer #4: All comments have been addressed

2. Does this manuscript meet PLOS Global Public Health’s publication criteria? Is the manuscript technically sound, and do the data support the conclusions? The manuscript must describe methodologically and ethically rigorous research with conclusions that are appropriately drawn based on the data presented.

Reviewer #3: Partly

Reviewer #4: Partly

3. Has the statistical analysis been performed appropriately and rigorously?

Reviewer #3: Yes

Reviewer #4: I don't know

4. Have the authors made all data underlying the findings in their manuscript fully available (please refer to the Data Availability Statement at the start of the manuscript PDF file)?

Reviewer #3: No

Reviewer #4: Yes

5. Is the manuscript presented in an intelligible fashion and written in standard English?

Reviewer #3: (No Response)

Reviewer #4: No

6. Review Comments to the Author

Reviewer #3: Thank for giving the chance to review the revised version such an important article that attempted to suggest integration as one strategy to mitigate the triplet disease in LMICs, mainly double burden addressed here. This review pick an important topic linking Infection disease prevention and control with NCD prevention and control in LMIC where a triple disease burden (accidents an injuries make the triplet) health set up is poor and resource constraints are there is very critical. Though the linkage by disease is narrow and has some limitations it will serve as stepping stone for future similar studies.

Reviewer #4: The background provides a solid foundation but could be enhanced by more clearly stating the specific objectives of the systematic review early on. A clearer articulation of how this review aims to fill existing knowledge gaps or address specific challenges in ID-NCD care integration would strengthen the introduction.:

The background introduces important concepts such as "syndemic disease" and "integrated ID-NCD health care." While these terms are crucial to the study's context, their definitions are briefly touched upon, which might leave readers unfamiliar with the subject matter, seeking more clarity. Expanding on these definitions, possibly with examples or a more detailed explanation of their relevance to the study, would enhance understanding and engagement. For instance, elaborating on how syndemic theory applies to the interaction between IDs and NCDs could illuminate the complexities of comorbidities and the systemic factors that exacerbate health outcomes.

Inclusion and Exclusion Criteria: clarifying the specifics regarding including studies up to December 2022 would enhance transparency.

Screening and Data Extraction Process: Calibration Exercise: A more detailed description of the calibration exercise, including how it was conducted, the criteria used for calibration, and the outcome measures, would be beneficial.

Is it possible to present the appraisal virtually? Additionally, how can we effectively present the findings using illustrations or visual aids?

7. PLOS authors have the option to publish the peer review history of their article (what does this mean?). If published, this will include your full peer review and any attached files.

**Do you want your identity to be public for this peer review?** For information about this choice, including consent withdrawal, please see our Privacy Policy.

Reviewer #3: No

Reviewer #4: No
